# Tertiary Lymphoid Structures (TLSs) and Stromal Blood Vessels Have Significant and Heterogeneous Impact on Recurrence, Lymphovascular and Perineural Invasion amongst Breast Cancer Molecular Subtypes

**DOI:** 10.3390/cells12081176

**Published:** 2023-04-17

**Authors:** Alina Cristina Barb, Mihaela Pasca Fenesan, Marilena Pirtea, Madalin Marius Margan, Larisa Tomescu, Eugen Melnic, Anca Maria Cimpean

**Affiliations:** 1Department of Microscopic Morphology/Histology, Victor Babes University of Medicine and Pharmacy, 300041 Timisoara, Romania; 2Doctoral School in Medicine, Victor Babes University of Medicine and Pharmacy, 300041 Timisoara, Romania; 3OncoHelp Hospital, 300239 Timisoara, Romania; 4Department of Functional Sciences, Discipline of Public Health, Victor Babes University of Medicine and Pharmacy, 300041 Timisoara, Romania; 5Department of Obstetrics and Gynecology, Victor Babes University of Medicine and Pharmacy, 300041 Timisoara, Romania; 6Department of Pathology, Nicolae Testemitanu State University of Medicine and Pharmacy, 2004 Chișinău, Moldova; 7Center of Expertise for Rare Vascular Disease in Children, Emergency Hospital for Children Louis Turcanu, 300011 Timisoara, Romania

**Keywords:** tertiary lymphoid structure, TLS, breast cancer molecular subtypes, immature blood vessels, mature blood vessels, recurrence, lymphovascular invasion, perineural invasion

## Abstract

**Simple Summary:**

Immunotherapies for breast cancer (BC) are among the most promising treatments for mammary gland malignancies, but their tissue and cellular targets are heterogeneous and controversial. Hence, the effectiveness of immunotherapy varies. Tertiary lymphoid structures (TLSs) are paid the most attention among these therapies due to the favorable prognostic function they have been shown to play for several malignancies, but this has less investigated for BC because they are not at all connected to some BC molecular subtypes and the stromal vascular network. Moreover, research on the TLS’s impact on lymphovascular, perineural, and BC recurrence is inconsistent. In the current investigation, significant variations between TLS positive and TLS negative subgroups within the same BC molecular subtype were observed and had a significant impact on BC recurrence, lymphovascular invasion, and perineural invasion. By promoting quick blood vessel maturation followed by lowering lymphovascular or perineural invasion, and recurrence, BC-associated TLSs seem to have a local protective role against tumor spread.

**Abstract:**

Background: Tertiary lymphoid structures (TLSs) mediate local antitumor immunity, and interest in them significantly increased since cancer immunotherapy was implemented. We examined TLS− tumor stromal blood vessel interplay for each breast cancer (BC) molecular subtype related to recurrence, lymphovascular invasion (LVI), and perineural invasion (PnI). Methods: TLSs were quantified on hematoxylin and eosin stain specimens followed by CD34/smooth muscle actin (SMA) double immunostaining for stromal blood vessel maturation assessment. Statistical analysis linked microscopy to recurrence, LVI, and PnI. Results: TLS negative (TLS−) subgroups in each BC molecular subtype (except to Luminal A) have higher LVI, PnI, and recurrence. A significant rise in LVI and PnI were observed for the HER2+/TLS− subgroup (*p* < 0.001). The triple negative breast cancer (TNBC)/TLS− subgroup had the highest recurrence and invasion risk which was also significantly related to tumor grade. PnI but not LVI significantly influenced recurrence in the TNBC/TLS+ subgroup (*p* < 0.001). TLS−stromal blood vessel interrelation was different amongst BC molecular subtypes. Conclusion: BC invasion and recurrence are strongly influenced by TLS presence and stromal blood vessels, especially for HER2 and TNBC BC molecular subtypes.

## 1. Introduction

Tertiary lymphoid structures (TLSs) are follicular-like lymphoid tissue accumulations inside other organs than lymphoid ones. TLSs are induced by various inflammatory [1,2], autoimmune [3], or malignant states of these tissues [4]. The TLS has subsequently been associated with inflammatory syndrome from various malignancies with an emphasis to their role as local, extra nodal antibody-generating structures developed in several non-lymphoid organs where they were described. 

Intratumorally, TLSs’ impact on patient’s prognosis and survival was first reported for early-stage non-small cell carcinoma (NSCC), but it has also been linked to TLS cellular heterogeneity composition described as a mixture of dendritic cells, CD4+ cells, and S100 beta+ cells [5]. Moreover, the same authors pointed out that TLS dendritic cells high in number may be considered an unfavorable prognostic factor [5]. Dendritic cell tumor infiltration was suggested as a potential prognostic criterion for stratifying early stage NSCC patients who might have an increased risk of recurrence [5]. Coppola and Mullem [6] highlighted TLSs as potential therapeutic targets based on immunological therapeutic strategies in malignant lesions.

Breast cancer (BC)-associated TLS presence and its prognostic role related to the benefits of antitumor responses has been controversial since the early description of these structures. Asport et al. [7] reported BC tissue infiltration by dendritic cells co-located with CD4+ T lymphocytes. They also observed that these infiltrating T-lymphocytes secreted IFN-gamma as well as elevated levels of IL-4 and IL-13, noting that malignant cells were highly positive for IL-13. This suggested a mutual relationship between IL-13-secreting T lymphocytes and BC malignant cells. Using an experimental model in mice, these same authors demonstrated that CD4+ T cells supported and stimulated early tumor development, which was shown to be dependent on dendritic cells and which could be partially inhibited by administration of IL-13 [8] antagonists.

The interest in TLSs significantly increased with the use of anti-PD-1 and -PD-L1 therapies as part of antineoplastic regimens [8,9]. TLSs’ prognostic and therapeutic role related to immunotherapy response in BC represents a relatively new research direction, with researchers mainly starting to focus on them since 2016. The novelty of the research topic as well as the extremely limited data regarding the impact of TLSs on BC prognosis and response to therapy is supported by the existence of a limited number of papers in the field [10].

TLSs have been reported in about 60% of breast cancers [11,12]. Most of the papers aiming at identifying TLSs’ existence or their impact on BC prognosis and therapy were mainly reported in the TNBC subtype [13], but the association of TLSs with other BC molecular subtypes is extremely limited [14,15].

The angiogenic switch is activated by an inflammatory millieu, which drives the development of new tumor stromal blood vessels. TLSs in breast cancer stroma are associated with tumor angiogenesis activation, suggesting that this may be a factor favoring BC metastasis [16], but most of these investigations were conducted in experimental models, with no data on human tissues [16].

Vascular Endothelial Growth Factor (VEGF) which is highly secreted under the influence of inflammation promotes stromal blood vessel endothelial cell activation followed by newly formed blood vessel development and their co-option by neighboring malignant tissues [17]. Tumor blood vessel endothelial cells (TECs) differ from normal ones in terms of phenotypic and functional features. The functional significance of TECs is likewise complicated. TLSs, which have recently been linked to the therapeutic response to checkpoint antibody therapy, are formed within the tumor thanks in part to TECs [18].

The TLS dynamic influenced by heterogeneous anti-tumor therapies is one of the most recent controversial issues related to malignancies therapy. Recently, low-dose stimulator of interferon genes (STING) delivery to tumor microenvironment (TME) from various cancers induced vascular normalization and stimulated TLS neogenesis [19,20,21]. Tumor-associated TLSs were associated with a good prognosis for most types of cancer. However, a small number of studies have found a link between TLSs and a bad prognosis [22].

The limited literature data described above related to TLSs’ different impacts on BC molecular subtypes and the TLS interplay with stromal vascular components determined the research direction of this study. We aim to study the interplay between TLSs and tumor stroma blood vessels (immature-CD34+/SMA−, versus mature-CD34+/SMA+), to find if this interrelation is BC subtype-specific and may influence lymphovascular and perineurial invasion and recurrence. 

## 2. Materials and Methods

### 2.1. Patients’ Selection and Ethical Issues

The present study is part of a large retrospective study including 150 breast cancer formalin fixed paraffin embedded (FFPE) specimens collected between 2015 and 2022 from women aged between 32 and 85 years old diagnosed with ductal invasive carcinoma on histopathology. A total of 53 cases out of 150 had a complete clinical, histopathological and therapeutic profile useful for our purpose. The selected clinic-pathologic and therapeutic parameters were age, menopausal status, BC molecular subtype, tumor grade (G), Nottingham Prognostic Index (NPI), Body Mass Index (BMI), lymphovascular/perineurial invasion, and local recurrence. At the time of diagnosis, no patients had a history of remotely certified imaging metastases. A molecular profile was assessed for each case by using immunohistochemistry markers such as estrogen receptor (ER), progesterone receptor (PR), proliferation index (Ki67), and human epidermal growth factor receptor 2 (HER2) to establish BC molecular subtypes. An informed consent was obtained from each patient based on a standardized form approved by the Research Ethics Committee of “Victor Babeș” University of Medicine and Pharmacy in Timișoara (No. 49/28 September 2018).

### 2.2. Primary Processing, Histopathology and FFPE Specimens’ Selection for Immunohistochemistry

Briefly, tumor fragments were collected by both mastectomy and needle core biopsy, for the initial diagnosis before any therapy. A representative portion of the tumor and the surrounding tumor area was chosen. Breast cancer samples were taken and preserved in buffered formalin for 24 to 48 h before being subjected to the standard paraffin embedding procedure. Three micrometers thick sections were obtained from each FFPE block and mounted on glass slides. Hematoxylin and eosin were used to stain one slide from each case for routine microscopic analysis. To validate the initial histopathology diagnosis and to assess tissue quality for a proper specimen’s selection for immunohistochemistry, two independent pathologists evaluated hematoxylin and eosin-stained slides, identified TLSs, and quantified them. Vimentin immunostaining was used to test the quality of the tissue (clone V9). Vimentin-positive tumor stroma cases were considered appropriate for selection and for the next immunohistochemistry procedures.

### 2.3. Immunohistochemistry

Immunohistochemistry (IHC) was performed by using Leica Bond-Max (Leica Biosystems, Newcastle upon Tyne, UK) Autostainer on 3 μm thick sections. Unmasking was carried out with Novocastra Bond Epitope Retrieval Solution 1 and 2, pH6 and 9 solutions (Leica Biosystems, Newcastle Ltd., Newcastle upon Tyne NE 12 8EW, UK). Inhibition of endogenous peroxidase was performed with 3% hydrogen peroxide for 5 min. A double immunostaining procedure was applied for a differential assessment of BC stroma mature and immature tumor blood vessels. We used CD34 mouse anti-human monoclonal antibodies (clone, QBEnd 10, Leica Biosystems, Newcastle upon Tyne, UK, 30 min incubation time at room temperature) for tumor vessel endothelium and smooth muscle actin (SMA) mouse anti-human monoclonal antibodies (clone 1A4, Leica Biosystems, Newcastle upon Tyne, UK, 30 min incubation time at room temperature) for muscle perivascular cells surrounding some stromal tumor blood vessels during their maturation process. Visualization systems as Bond Polymer Refine Detection System DAB and the Bond Polymer Refine Red Detection System were the following steps for completing the immunohistochemical procedure. CV Mount (mounting medium from Leica Biosystems, Newcastle Ltd., UK) was used as a permanent mounting medium for immunohistochemistry-stained slides. CD34/SMA double stain protocol for automated immunohistochemistry was a modified version of previous similar staining manually or automated performed in our lab [23,24] and was adapted for Leica Bond-Max Autostainer detection kits.

### 2.4. TLS Assessment and IHC Specimens’ Analysis

Hematoxylin and eosin stained and IHC slides were scanned by using Grundium OCUS 20 Microscope (Grundium, Tampere, Finland) and archived as svs. format in the Case Center Slide Library (3DHistech, Budapest, Hungary). From this digital library the slides were uploaded on QuPath version 0.4.2., an open-source platform for bioimage analysis of microscopic slides where they were analyzed by using integrated software and its extensions as Fiji and Vascular Analysis. For each case we selected three stromal areas with the highest density of stromal blood vessels close to tumor areas and TLSs (where they are present) by using QuPath annotation tools. We differentially counted at X400 magnification the mature (CD34+/SMA+) and immature (CD34+/SMA−) in the same selected area. An average of mature and immature stromal blood vessels from those three areas were used for our purpose. 

TLS assessment on H&E was performed according to microscopic features of tertiary lymphoid structures previously described in several papers [25,26,27]. Based on these previous descriptions, round or oval well-defined lymphoid tissue organized in structures mimicking lymphoid follicles from lymph nodes inside TME or BC tumor tissue were considered tertiary lymphoid structures and were quantified regarding their presence and number.

### 2.5. Statistical Analysis, Correlation with Clinical and Therapeutic Data

JAMOVI software for macOS devices was used for statistical analysis. The presence of TLSs was correlated to BC molecular subtypes, menopausal status, immature and mature stromal blood vessels, and recurrence of lymphovascular and perineural invasion. A statistical correlation was assessed and considered significant for a *p* value of 0.05 or less.

## 3. Results

### 3.1. Histopathology Assessment, BC Cases Molecular Classification, and TLSs

The 53 cases of malignant breast tumors included in this study were microscopically diagnosed as invasive ductal breast carcinomas. A total of 69.82% out of the all cases had postmenopausal status. Cases were evaluated for tumor differentiation grade (G) and the Nottingham Prognostic Index (NPI). Of the total number of breast carcinomas included in this study, 3.77% were evaluated as G1, 75.4% were evaluated as G2, and 20.86% as G3. Regarding the NPI, 1.88% of cases had an NPI of 4, 3.77% had an NPI of 5, 35.4% had an NPI of 6, 73% had an NPI of 7, and 20.75% had an NPI of 8. BC molecular subtype analysis based on ER, PR, HER2, and Ki67 markers revealed that 16.98% were Luminal type A (LA), 35.85% were Luminal type B (LB), 13.2% were LB-HER2+ type, 5.66% were HER2+ type, and 28.3% were triple negative breast cancer (TNBC) type. TLSs were detected in 54.71% of the total cases. From this TLS positive group, 86.95% were postmenopausal women while 13.05% were patients before menopause. In the TLS positive subgroup, case distribution according to molecular type was as follows: 24.13% of cases were of LA type, 27.58% were LB type, 10.34% were LB-HER2 type, and 37.93% were TNBC type. The TLS negative subgroup included 8.33% LA type, 45.83% LB type, 16.67% LB-HER2 type, 12.5% HER2 type, and 16.66% TNBC type.

### 3.2. TLS Classification and Their Interrelation to Tumor Stroma Blood Vessels, Adipose Tissue, and BMI

TLSs were not detected inside the normal breast tissues close to tumor tissue (Figure 1A). We observed a high homogeneous TLS morphology for all BC molecular subtypes. The number of TLSs varied between one and three per case. Round or oval well-organized lymphoid follicle-like structures (Figure 1B) were detected in most (90%) of the 53 cases heterogeneously distributed inside the TME (Figure 1C).

About 5% out of the 53 BC cases had intratumor TLSs (Figure 1G,J) while the remaining 5% of the 53 cases showed TLSs in the BC tumor microenvironment far from the tumor often in the stroma and the adipose tissue surrounding the mammary gland (Figure 1D,E). The intratumor TLSs are smallest in size (Figure 1G,J) but are heterogeneous regarding density and blood vessel types. TLSs close to the adipose tissue were big in size and usually not sharply delineated from the surrounding tissue (Figure 1E). Two types of TLSs related to adipose tissue were detected: (i) in the tumor stroma and the adipose tissue (Figure 1D,I) and (ii) in the tumor cell area and the surrounding adipose tissue (Figure 1E). This TLS type was highly vascularized compared to the intratumor TLSs or those found in normal breast tissue and tumor area (Figure 1B). The morphology and phenotype of intra-TLS vessels were highly suggestive of an active angiogenic process, and their lumen was lined by high endothelial cells similar to those from regular lymph nodes. This is most probably due to the dual adipose tissue’s ability to induce both inflammation (with development of TLSs) and angiogenesis (certified in our study by the presence of intra-TLSs of immature CD34+/SMA− and mature CD34+/SMA+ blood vessels with a particular and heterogeneous morphology highly suggestive of an active angiogenic process). A high angiogenic process may be a factor favoring metastases. In light of this, we assessed TLS presence related to BMI to find if normal, overweight, or obese status may influence TLS presence in BC stroma. A significant correlation was found between BMI and TLS presence (*p* = 0.014, Figure 2a). The next step of evaluation was to find if there is a correlation between TLS and blood vessel type (immature versus mature).

As is shown in Figure 2b, TLS presence was correlated with both immature IBV_CD34+/SMA− (*p* = 0.008) and mature MBV_CD34+/SMA+ (*p* = 0.003) stromal blood vessel density when we applied a global assessment for all molecular subtypes.

### 3.3. TLS, BC Molecular Subtypes, and Stromal Tumor Blood Vessels

TLS percentual distribution heterogeneity previously observed amongst BC molecular subtypes represented our starting point for continuing the study of TLS interrelation with tumor stroma blood vessels type and BMI specific for each molecular subtype. No data about TLS related to Luminal A, B, or Luminal B-HER2 molecular subtypes are available at the moment nor about their relationship to BMI, age, recurrence, or lymphovascular and perineural invasion. For each molecular subtype we performed a global analysis (for both TLS+ and TLS− cases) and a specific analysis separating TLS+ from TLS− in the same molecular subtype.

Global analysis for Luminal A-BC cases showed no significant correlations between TLS and any parameters chosen for the present study (BMI, age, immature versus mature blood vessels, NPI, tumor grade, lymphovascular invasion, perineurial invasion, or recurrence rate) excepting an inverse, low and partially significant correlation between TLS and age (*p* = 0.045). All patients aged between 42 and 70 years old presented with TLSs while for the patients over 70 years old we were not able to detect stromal TLSs. Despite the fact that there were not found any direct correlations between TLSs, IMBV_CD34+/SMA−, and BMI, we considered it important to mention that overweight or obese patients (BMI over 24.9) with Luminal A-BC had the highest IMBV_CD34+/SMA− vessel density, and this was statistically significant (*p* = 0.029). This finding suggests that adipose tissue may promote the formation of new stromal blood vessels during Luminal A-BC progression independently by TLS presence. When we assessed the TLS+_Luminal A-BC cases subgroup, we found that BMI was significantly correlated to IMBV_CD34+/SMA− tumor stroma vessel density (*p* = 0.015). For the TLS−_Luminal A subgroup, no significant correlations were found.

The Luminal B-BC subgroup was one of the two BC molecular subtypes (together with the TNBC-BC subgroup) with a high percentage of TLS positive cases. Global analysis of both TLS+ and TLS− Luminal B-BC cases revealed significant correlations between TLS, IMBV_CD34+/SMA− (*p* = 0.017), and MBV_CD34+/SMA+ stromal blood vessels (*p* = 0.042). In the TLS positive Luminal B-BC subgroup, patients of younger age developed a higher number of IMBV_CD34+/SMA− stromal blood vessels (*p* = 0.017) compared to elderly patients. No significant influence of TLS presence was found for lymphovascular invasion, perineurial invasion, or recurrence related to both types of tumor stroma blood vessels. The TLS negative group was characterized by different data compared to TLS positive group. Lack of TLSs induced a significant perineurial invasion for elderly patients (*p* = 0.024) being strongly correlated to postmenopausal status (*p* = 0.019). Perineurial invasion was significantly associated with lymphovascular invasion for TLS negative patients (*p* = 0.019) but not with recurrence rate (*p* = 0.695). Recurrence rate was highly influenced by NPI (*p* = 0.008) and tumor grade (*p* = 0.024) but not by lymphovascular or perineural invasion. No significant BMI impact was detected for the Luminal B-BC subgroup related to TLSs and stromal blood vessels.

For the TLS negative Luminal B-HER2 subgroup the most interesting finding was a significant correlation of recurrence with both perineurial (*p* < 0.001) and lymphovascular invasion (*p* < 0.001). A significant correlation between NPI and menopausal status for this group was found (*p* = 0.038). In the TLS positive Luminal B-HER2 subgroup, we did not detect any cases with recurrence, or perineurial or lymphovascular invasion. No significant or relevant findings had been observed related to both types of stromal blood vessels.

All HER2 positive BC cases were negative for TLSs. The absence of TLSs favored the development of IMBV_CD34+/SMA−. This is supported by our finding of a high IMBV_CD34+/SMA− vessel density but also by the strong significant inverse correlation found between IMBV_CD34+/SMA− and lymphovascular invasion (*p* < 0.001), perineurial invasion (*p* < 0.001), and recurrence (*p* < 0.001). For young patients, the recurrence rate was significantly higher compared to elderly patients (*p* < 0.001) and was strongly correlated to lymphovascular (*p* < 0.001) and perineurial invasion (*p* < 0.001).

TNBC-BC cases represent a controversial group related to prognosis, therapy response, and overall survival. Our study revealed that the TNBC-BC subgroup included the highest percentage of TLS+ cases. The most interesting data related to the overall assessment of TNBC-BC cases were related to recurrence, and lymphovascular and perineurial invasion. For the TLS+_TNBC subgroup, recurrence was strongly correlated to perineurial invasion (*p* < 0.001) but not to lymphovascular invasion (*p* = 0.104). It seems that perineurial invasion is strongly influenced by MBV_CD34+/SMA+ density by an inverse correlation (*p* = 0.026). Tumor stroma blood vessel maturation followed by the increase in MBV_CD34+/SMA+ density induced a low perineurial invasion. The TLS−_TNBC subgroup was unique between all BC molecular subtypes because of a significant correlation found for lymphovascular invasion with tumor grade (*p* < 0.001). A similar strong interrelation was observed between recurrence to tumor grade (*p* < 0.001), and lymphovascular (*p* < 0.001) and perineurial invasion (*p* < 0.001) but not with NPI. 

The different TLSs’ impact amongst BC molecular subtypes related to NPI, G, recurrence, LVI, and PnI is summarized in Table 1. 

Based on summarized data from Table 1, we report three main important findings: (i) the TLS negative subgroups are strongly associated with an increased lymphovascular or perineural invasion and some of them with a high recurrence rate except the Luminal A type; (ii) the TLS negative subgroups associating HER2 oncoprotein (including Luminal B_HER2 and HER2) have similar data supporting the increase in recurrence rate closely related to lymphovascular and perineural invasion; (iii) the TNBC subgroup remains the highest risk subgroup related to TLS impact on recurrence and invasion. Compared to other TLS negative subgroups, TNBC-TLS negative subgroup recurrence is not related just to lymphovascular and perineural invasion but also to G. Additionally, tumor differentiation seems to strongly influence both perineural and lymphovascular invasion for the TNBC-TLS negative subgroup.

## 4. Discussion

TLSs’ involvement in malignancies is not a new concept in the field [1,4,6]. Despite this, not much attention has been paid to TLSs until cancer immunotherapy was implemented as one of the therapeutic strategies for malignant diseases [25,26,27].

From the first, unique paper, for that year (1952), by Peyton Rous and his collaborators, which proved in an experimental model that the immune microenvironment plays an important role in tumor progression [28], to the first two months of the present year (2023), when more than 3000 papers have already published in the field of cancer immunotherapy [29], the interest in TLSs have continuously increased. Despite the evidence, several controversies persist related to the cellular and molecular features of the tumor immune microenvironment, including TLSs [25,30]. The interactions between the immune component and other TME components such as stromal blood vessels or tumor stromal myofibroblasts are still less elucidated [31,32,33,34]. The TLSs impact on tumor progression, lymphovascular invasion, perineural invasion, and recurrence remain the most questionable and neglected chapters in the field [35,36,37,38,39].

TLSs were studied predominantly in TNBC and HER2-BC molecular subtypes [40,41,42,43]. Most of the papers studied intratumor TLSs while stromal TLSs are not very well characterized regarding their topography and interrelation to peritumor adipose tissue.

Together with intratumor TLSs we described two different types of stromal TLSs characterized by their proximity to peritumor adipose tissue and tumor cells. We found this observation interesting considering that adipose tissue has both well certified proinflammatory and angiogenic properties in several tumor types but not in breast cancer [44,45,46]. Associations between TLS formation and adipocyte have been described for Crohn’s disease but not for cancer [47]. A significant correlation found between BMI and TLSs in the present study, represents the first evidence of adipose tissue influence on TLS development in breast cancer and suggests that BC-peritumor adipose tissue may have a role in the development of TLSs. In a few relatively recent studies in the field of malignant diseases, BMI has been proven to strongly influence the immune tumor microenvironment and intratumor angiogenesis. Intriguingly, in a paper by Sanchez and *colab.* [46] it was reported that in renal cell carcinoma (RCC), obese patients showed a higher tumor immune infiltration and increased peritumor adipose tissue inflammation compared to normal weight patients. Additionally, they pointed out that tumor microenvironment variability related peritumor adipose tissue is highly dependent on BMI, and RCC obese patients with high immune infiltration of peritumor adipose tissue had better overall survival compared to normal weight patients. The authors of the previously mentioned study referred to diffuse immune infiltration but not to TLSs [46]. In the present study we proved that BMI significantly influenced TLS presence in some subtypes of BC, specifically for the Luminal A subtype. Being the first study in the BC field which mentioned this correlation, we suggest that further studies will be needed for a more accurate analysis of the TLS and adipose tissue interrelation. Additionally, the TLS distribution between tumor stroma or tumor cells and adipose tissue described in the present study supports the hypothesis that TLSs may be developed as a local defense mechanism against loco-regional tumor spread and metastases and encourage development of further studies in the field.

It is well known that tumor infiltrating lymphoid tissue induces a high angiogenic response certified by the increase in newly formed blood vessel density inside the tumor area and tumor stroma [48] and by vessel heterogeneity regarding their immature active status or their full maturation by acquisition of perivascular smooth muscle actin (SMA) positive cells. Lymphoid tissue is the main component of TLSs, but the impact of TLSs on stromal tumor angiogenesis was not paradoxically assessed previously. High endothelial venules inside TLSs were reported to be one of the main contributors of maintaining chronic inflammation in cancer by recirculating immune cells [49,50]. Sawada et al. [51] performed comparative transcriptome analysis of human breast cancer to investigate genes differentially expressed between tumor associated HEVs and the rest of the tumor vasculature. They isolated HEVs from BC-TLS-rich areas and from BC-TLS-free areas by laser capture microdissection and compared the gene expression profiles. They found a specific gene signature for HEVs, and they proved that high transcript counts of HEV signature genes from BC-TLS-rich areas are strongly associated with prolonged breast cancer survival. They strongly recommended TLS detection for BC patients as a routine procedure during microscopic evaluation [51] followed by gene signature analysis to predict patients’ prognoses.

In the same paper, the authors mentioned another interesting finding related to the gene expression profile heterogeneity of TLS-associated blood vessels [51]. We also detected a blood vessel heterogeneity in BC tumor stroma containing a mixture of immature blood vessels with a discontinuous wall (IMBV_CD34+/SMA−, known to be one of the disseminating routes for malignant cells) and mature blood vessels (MBV_CD34+/SMA+, which are stable vessels with silent, inactive endothelial cells and a well-defined wall mimicking normal blood vessels). In this study, we certified the angiogenic and immunohistochemical heterogeneity of tumor stromal vessels associated with TLSs. Through the statistically significant correlation found in the present study between TLSs and both types of stromal vessels we certify the TLS’s dual role of sustaining an active angiogenic process counterbalanced by a similar rate of newly formed blood vessel maturation. 

The TLS crosstalk to stromal vasculature seems to be modulated by endothelial cell specific gene signatures, most probably due to Notch signaling pathway involvement [31]. Current data support the theory that endothelial cell gene expression signature associated with TLSs seems to be organ specific [16] as has been stated for RCC [44] and breast cancer [51]. 

There is only one published paper [52] where TLS presence was evaluated separately for each molecular subtype, but it lacks Luminal B-HER2 type from the analysis. 

The present study differentially assessed for the first time TLS positivity percentage for all BC molecular subtypes. Luminal B BC-subtype had the highest percentage (27.58%) of TLS positive cases amongst Luminal subtypes followed by Luminal A-BC (24.13%), and Luminal B-HER2 (10.34%). Our Luminal A-TLS + BC cases percentage was relatively close to that reported by Liu et al. [52] (24.13% in the present study versus 21.23% in the previous one) but for other molecular subtypes Liu et al. reported higher TLS + BC cases (49.12% for Luminal B, 56.66% for HER2, and 48.8% for TNBC). The lowest percentage of TLS + cases was recorded for Luminal B-HER2 BC cases, and this together with a lack of TLS + HER2 subtype cases suggested a potential negative effect of HER2 overexpression on TLS development. This intriguing finding forced us to check the literature in the field of BC related to TLS reports on HER2 positive BC cases. We were surprised to find out that all papers related to HER2 + BC referred to tumor diffuse infiltrating lymphocyte (known as TIL) but not to tertiary lymphoid structures (TLSs, defined as lymphoid follicle-like structures developed in a non-lymphoid tissue) which are the aim of our study [53]. Based on this evidence we consider our findings to be correct until new data is available. 

The TLS + cases highest percentage was reported for the TNBC-BC subgroup in the present study. Triple negative breast cancer (TNBC) is one of the most aggressive BC subtypes with a less efficient therapy, and poor prognosis and survival [54,55,56]. Jézéquel et al. [55] recently described three distinct TNBC clusters named C1, C2, and C3, based on gene expression signatures and TLSs and TIL microscopic assessment. The authors reported a very interesting finding for the TNBC C1 cluster. Despite of the fact that their TNBC cases were declared as being negative for HER2 oncoprotein by immunohistochemistry (according to interpretation provided by the manufacturer), the *ERBB2* gene expression signature for the C1 cluster displayed a high expression of the *ERBB2* pathway [54]. Their observation partially overlapped to ours. Based on the descriptive criteria, in both their study and ours, the C1 cluster was found to be overlapping with TLS negative TNBC cases. In the present study, about half of TNBC cases were scored with 1 in IHC analysis and thus, they were considered *HER2* negative. We found that both the HER2 positive subtype (which all lacked TLSs) and the TLS negative TNBC subtype showed similar, significant strong correlations between TLS lack, recurrence, and perineural and lymphovascular invasion not found for other molecular subtypes such as Luminal A or Luminal B cases. 

In the same paper [54], the authors also reported that the TNBC C2 cluster lacks TLSs but has tumor infiltrating lymphocytes (TILs) with a pro-tumorigenic immune response (immune suppressive), high neurogenesis (nerve infiltration), and high biological aggressiveness [54]. We supposed that this cluster may correspond to our TLS negative TNBC subtype based on the strong correlations found not only between recurrence and perineural invasion but also to lymphovascular invasion. The TLS negative TNBC subgroup from our study was the only group where tumor grade was significantly correlated with recurrence, and lymphovascular and perineural invasion, suggesting more evidence of the biological aggressiveness previously described for the C2 cluster.

One of the main C3 cluster features mentioned by the authors was TLS presence and their cellular composition influence on this TNBC subgroup’s prognosis and therapy response [54]. C2 and C3 clusters are TNBC subgroups where immunotherapies should be applied but in a different manner for each; as immune stimulation for C2 and immunomodulatory activity for C3 [54]. Thus, the authors recommended TLS assessment as a mandatory step of BC FFPE specimens’ histopathologic evaluation and their inclusion in prognostic criteria for recurrence and invasion. 

Last, but not the least, more attention should be paid to the TLS’s influence on tumor stroma immature and mature blood vessel development. Surprisingly, the TLS role on surrounding tumor stroma blood vessels was completely neglected, with the TLS interrelation to blood vessels being limited to the high endothelial vessel assessment as part of the criteria used to identify TLSs microscopically [57,58]. Diffuse inflammatory cell infiltration (currently defined as TIL) into malignant tissue stroma induced a high angiogenic response, which is responsible for an increased tumor stroma blood vessel density associated with a high rate of lymphovascular invasion and metastases [58,59,60,61]. Our search on PubMed and other databases did not help us to find any information related to interactions between TLSs, immature and mature stromal blood vessels, or tumor angiogenesis for any BC molecular subtype. Thus, we consider this paper the first report suggesting a potential impact of TLSs differentially assessed for immature and mature stromal blood vessels in different BC molecular subtypes. Global analysis showed a significant direct correlation between both mature (MBV_CD34+/SMA+) and immature (IMBV_CD34+/SMA−) blood vessels. It is widely accepted that lymphovascular invasion is used as dissemination route for immature tumor vessels with discontinuous walls and lacking SMA positive perivascular cells. Starting from this observation, we continued the study for each BC molecular subtype, and found that TLS presence is significantly associated with MBV_CD34+/SMA+ for Luminal B and TNBC-BC subtypes, suggesting TLS involvement in a more rapid maturation (developed by SMA positive perivascular smooth muscle cell acquisition) of BC stromal blood vessels. This hypothesis may be partially sustained by the lack of significant correlations observed between recurrence and lymphovascular invasion for TLS+_Luminal B and TLS+_TNBC subgroups from the present study and the presence of strong correlations between recurrence and lymphovascular invasion found for TLS−_Luminal B and TLS−_TNBC subgroups. Only one recent study linked TLSs to tumor angiogenesis by the assessment of HEVs and angiogenesis inside TLSs from colon cancer [62], and the authors briefly stated that less neovascularization around TLSs happens most probably due to the inhibitory effects of CD8+ T cells on endothelial cell proliferation and angiogenesis [63]. 

The mechanistic investigation of TLSs at the cellular or molecular levels, however, is still in the early stages. An overview on mechanistic and therapeutic studies related to TLS research benefits is presented in Table 2, which offers us a global evaluation of previous studies and helps to critically discuss the main findings of the present paper.

## 5. Conclusions

In the current investigation, significant variations between TLS+ and TLS− subgroups within the same BC molecular subtype were observed as having a significant impact on BC recurrence, LVI, and PnI. By promoting quick blood vessel maturation followed by lowering lymphovascular or perineural invasion and recurrence, BC-associated TLS seems to have a local protective role against tumor spread. TNBC/TLS− and HER2+/TLS− (HER2 and Luminal B_HER2) subgroups are identified to be highly aggressive subgroups where a high recurrence rate seems to emerge from a crosstalk between tumor and stromal components. TLS impact on stromal angiogenesis needs further validation by the assessment of cellular pathways influencing vessel proliferation and maturation.

## Figures and Tables

**Figure 1 cells-12-01176-f001:**
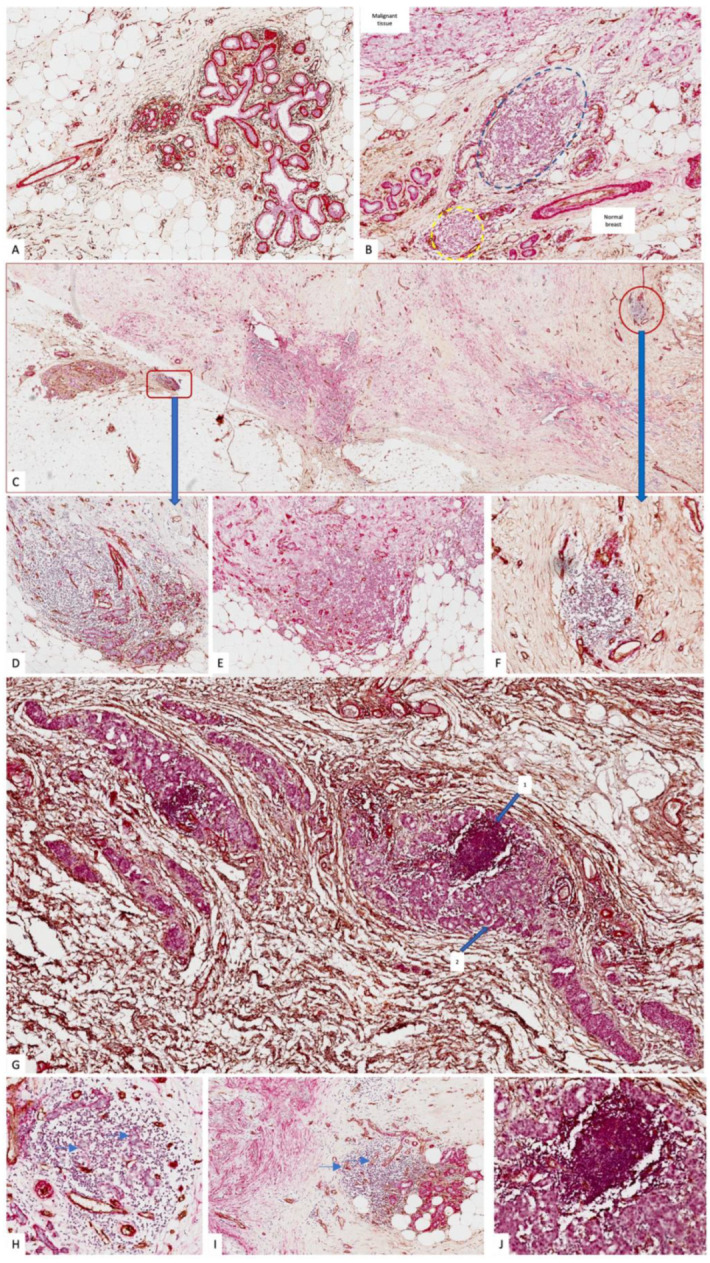
(**A**–**J**) Comparative TLS assessment distribution from normal to malignant breast cancer tissue. (**A**) No TLSs were detected inside normal breast tissue. (**B**) TLS complex interrelations with tumor tissue (upper left) but also with several TME components such as stromal blood vessels and adipose tissue. (**C**) General overview of TLS topography in mammary gland stroma related to its neighborhood with adipose tissue. (**D**,**E**) TLSs in direct contact with adipose tissue or (**F**) surrounded by stroma. (**G**,**J**) Intratumor TLSs are smaller in size ((**G**), inset 1) and surrounded by tumor cells ((**G**), inset 2). (**H**) Sometimes, TLSs are invaded by tumor cells (arrows) in a manner mimicking lymph node metastasis or (**I**) are highly vascularized (arrows) without any tumor cells inside. (**A**–**J**) Double stain for CD34 positive reaction for endothelial cells highlighted in brown with diaminobenzidine and smooth muscle actin (SMA) for perivascular cells highlighted in red with permanent AEC chromogen.

**Figure 2 cells-12-01176-f002:**
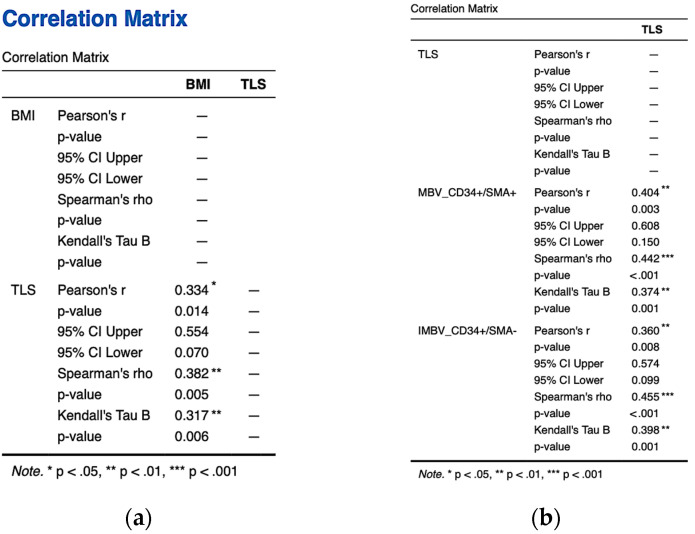
Correlation matrix showing the interrelation between BMI and TLS stromal blood vessels in BC tissues (*p* = 0.014 was significant). BMI significantly influenced the TLS presence (**a**) while TLSs are highly correlated with both immature (IBV_CD34+/SMA−, *p* = 0.008) and mature tumor (MBV_CD34+/SMA+, *p* = 0.003) stroma blood vessels (**b**).

**Table 1 cells-12-01176-t001:** Significant correlations between TLS− and TLS+ subgroups from each molecular subtype to G, NPI, LVI, PnI, and recurrence.

*BC Molecular Subtype*	Parameters	G	NPI	Recurrence	LVI	PnI
**TLS+_LUMINAL A**	G	*-*	** *p = 0.004* **	*NS*	*NS*	*NS*
	NPI	** *p = 0.004* **	*-*	*NS*	*NS*	*NS*
	RECURRENCE	*NS*	*NS*	*-*	*NS*	*NS*
	LVI	*NS*	*NS*	*NS*	*-*	** *p < 0.001* **
	PnI	*NS*	*NS*	*NS*	** *p < 0.001* **	*-*
**TLS−_LUMINAL A**	G	*-*	*NS*	*NS*	*NS*	*NS*
	NPI	*NS*	*-*	*NS*	*NS*	*NS*
	RECURRENCE	*NS*	*NS*	*-*	*NS*	*NS*
	LVI	*NS*	*NS*	*NS*	*-*	*NS*
	PnI	*NS*	*NS*	*NS*	*NS*	*-*
**TLS+_LUMINAL B**	G	*-*	** *p = 0.024* **	*NS*	*NS*	*NS*
	NPI	** *p = 0.024* **	*-*	*NS*	*NS*	*NS*
	RECURRENCE	*NS*	*NS*	*-*	*NS*	*NS*
	LVI	*NS*	*NS*	*NS*	*-*	*NS*
	PnI	*NS*	*NS*	*NS*	*NS*	*-*
**TLS−_LUMINAL B**	G	*-*	** *p = 0.008* **	** *p = 0.024* **	*NS*	*NS*
	NPI	** *p = 0.008* **	*-*	** *p = 0.008* **	*NS*	*NS*
	RECURRENCE	** *p = 0.024* **	** *p = 0.008* **	*-*	*NS*	*NS*
	LVI	*NS*	*NS*	*NS*	*-*	** *p = 0.019* **
	PnI	*NS*	*NS*	*NS*	** *p = 0.019* **	*-*
**TLS+_LUMINAL B-HER2**	G	*-*	*NS*	*NS*	*NS*	*NS*
	NPI	*NS*	*-*	*NS*	*NS*	*NS*
	RECURRENCE	*NS*	*NS*	*-*	*NS*	*NS*
	LVI	*NS*	*NS*	*NS*	*-*	
	PnI	*NS*	*NS*	*NS*	*NS*	*-*
**TLS−_LUMINAL B-HER2**	G	*-*	*NS*	*NS*	*NS*	*NS*
	NPI	*NS*	*-*	*NS*	*NS*	*NS*
	RECURRENCE	*NS*	*NS*	*-*	** *p < 0.001* **	** *p < 0.001* **
	LVI	*NS*	*NS*	** *p < 0.001* **	*-*	** *p < 0.001* **
	PnI	*NS*	*NS*	** *p < 0.001* **	** *p < 0.001* **	*-*
**TLS+_HER2**	G	*-*	*-*	*-*	*-*	*-*
	NPI	*-*	*-*	*-*	*-*	*-*
	RECURRENCE	*-*	*-*	*-*	*-*	*-*
	LVI	*-*	*-*	*-*	*-*	*-*
	PnI	*-*	*-*	*-*	*-*	*-*
**TLS−_HER2**	G	*-*	*NS*	*NS*	*NS*	*NS*
	NPI	*NS*	*-*	*NS*	*NS*	*NS*
	RECURRENCE	*NS*	*NS*	*-*	** *p < 0.001* **	** *p < 0.001* **
	LVI	*NS*	*NS*	** *p < 0.001* **	*-*	** *p < 0.001* **
	PnI	*NS*	*NS*	** *p < 0.001* **	** *p < 0.001* **	*-*
**TLS+_TNBC**	G	*-*	** *p = 0.042* **	*NS*	*NS*	*NS*
	NPI	** *p = 0.042* **	*-*	*NS*	*NS*	*NS*
	RECURRENCE	*NS*	*NS*	*-*	*NS*	** *p < 0.001* **
	LVI	*NS*	*NS*	*NS*	*-*	*NS*
	PnI	*NS*	*NS*	** *p < 0.001* **	*NS*	*-*
**TLS−_TNBC**	G	*-*	*NS*	** *p < 0.001* **	** *p < 0.001* **	** *p < 0.001* **
	NPI	*NS*	*-*	*NS*	*NS*	*NS*
	RECURRENCE	** *p < 0.001* **	*NS*	*-*	** *p < 0.001* **	** *p < 0.001* **
	LVI	** *p < 0.001* **	*NS*	** *p < 0.001* **	*-*	** *p < 0.001* **
	PnI	** *p < 0.001* **	*NS*	** *p < 0.001* **	** *p < 0.001* **	*-*

Legend: TLS—tertiary lymphoid structure; G-tumor grade; NPI—Nottingham Prognostic Index; LVI—lymphovascular invasion; PnI—perineural invasion; NS—non-significant correlation.

**Table 2 cells-12-01176-t002:** Critical TLS overview impact of the most relevant previous findings related to the results of the present study.

Study Title and Publication Year	Main Findings	Present Study Related Findings	Critical Discussions	Ref.
Tertiary lymphoid structures in breast ductal carcinoma in situ correlate with adverse pathological parameters. Zeng et al., 2023	TLSs in DCIS were associated with unfavorable prognostic features. No significant associations between TLSs and recurrence.	TLSs were assessed in invasive ductal carcinomas classified according to molecular profile. TLSs were significantly correlated with recurrences depending on BC molecular subtypes.	Recurrence rate increased with BC progression from an in situ to invasive state highly dependent on molecular subtypes.	[39]
The Presence of Tertiary Lymphoid Structures Provides New Insight Into the Clinicopathological Features and Prognosis of Patients With Breast Cancer. Wang et al., 2022	A review based on 15 papers from the literature. HER2 high expression was associated with high TLS presence.	All HER2 positive cases did not present TLSs and had significant correlation of recurrence with LVI, PnI, and young age but also with immature stromal blood vessels. HER2 positivity association to Luminal B (LB-HER2 subtype) showed similar findings with HER2 subtype.	Contradictory findings between these two studies support further research development to elucidate HER2 positivity impact on TLSs dynamic in BC. TLSs assessment on LB-HER2 subtype may be considered a novelty of the present study together with associated stromal blood vessels assessment.	[40]
Multicenter phase II trial of Camrelizumab combined with Apatinib and Eribulin in heavily pretreated patients with advanced triple-negative breast cancer. Liu et al., 2022	Camrelizumab plus apatinib and eribulin induced a better objective response rate and a higher progression free survival for TLS positive TNBC patients.	Lack of TLSs in TNBC was significantly correlated with lymphovascular and perineural invasion, and recurrence. TLS positive TNBC has mature blood vessels which induce a low perineural invasion	High rate of lymphovascular/perineural invasion and high recurrence related to TLSs may select TNBC patients who may have a low objective rate response to therapy and a lower progression free survival. Mature blood vessels are the proper route for drug delivery, and thus, their presence may be evaluated before therapy. TLS_stromal blood vessels type has not been previously studied.	[62]
Therapeutic Induction of Tertiary Lymphoid Structures in Cancer Through Stromal Remodeling. Johansson & Ganss, 2021	Insights into TLS development in normal and malignant conditions. TLSs in pancreatic cancer is associated with a more mature vascular network suggesting a possible link between TLS formation and stabilized tumor vessels.	TLS presence was significantly correlated with mature blood vessels surrounded by perivascular SMA positive cells which lack correlation with lymphovascular invasion.	Present study results related to TLSs and blood vessel maturation proved interplay between TLSs and the BC stromal vascular network in breast malignant tissue not found in other BC studies. Our findings are preliminary and supported by similar findings from other cancer types such as pancreatic cancer. Further studies are needed for validating these preliminary data.	[64]

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
