# Peer review of "Tertiary Lymphoid Structures (TLSs) and Stromal Blood Vessels Have Significant and Heterogeneous Impact on Recurrence, Lymphovascular and Perineural Invasion amongst Breast Cancer Molecular Subtypes"

_cells, 2023, doi:10.3390/cells12081176_

Round 1
Reviewer 1 Report
The manuscript- entitled Tertiary Lymphoid Structures (TLS) and stromal blood vessels 2 have significant and heterogeneous impact on recurrence, lym-3 phovascular and perineural invasion amongst breast cancer 4 molecular subtypes by Alina et al., aims to assess the TLS 23 and tumor stromal blood vessels for each BC molecular subtypes related to recurrence, lymphovas-24 cular (LIV) and perineural invasion (PnI).
I read the manuscript, and it is largely a well-planned, scientifically well-executed, and interesting work which is useful to investigators working in the area of breast cancer.
I have a few minor observations and recommend its acceptance.
• The abstract is too long and needs to be revised for conciseness.
• Authors must cite the references from where the protocol for performing experiments was adapted or adapted with modification from some source.
• Fig 1 needs to be improved for quality
Author Response
Dear Reviewer,
We would like to thank you for your time given for reviewing our paper. Your comments were valuable for us and helped us to improve the quality of the submission.
Please, find below our answers point by point to your comments. According to the instructions received from the editor, both for this letter and manuscript all the revisions will be highlighted in red.
With all our gratitude, in the name of manuscript authors,
Anca Maria Cimpean, MD, PhD, Hab.Dr., Full Professor of Histology, Pathologist
The manuscript- entitled Tertiary Lymphoid Structures (TLS) and stromal blood vessels 2 have significant and heterogeneous impact on recurrence, lymphovascular and perineural invasion amongst breast cancer 4 molecular subtypes by Alina et al., aims to assess the TLS and tumor stromal blood vessels for each BC molecular subtypes related to recurrence, lymphovascular (LIV) and perineural invasion (PnI).
I read the manuscript, and it is largely a well-planned, scientifically well-executed, and interesting work which is useful to investigators working in the area of breast cancer.
Thank you for your kind appreciation.
I have a few minor observations and recommend its acceptance.
- The abstract is too long and needs to be revised for conciseness.
The abstract had 196 words and is in concordance with the Instructions for Authors recommended by the journal (abstract up to 200 words). But you are right, it was too long, and we follow your recommendation to short it and make it more concise. You will find the new version of the abstract in red in the manuscript. Now, the new abstract version has about 160 words and it was extensively revised for conciseness by removing non-relevant data
- Authors must cite the references from where the protocol for performing experiments was adapted or adapted with modification from some source.
The immunohistochemistry protocol applied for the present study was standardized in our Immunohistochemistry Lab in Timisoara (by me, about 12 years ago). I did really appreciate your suggestion to cite our protocol in this paper. Thank you, again! (Ferician and Gurzu, please see in the text)
- Fig 1 needs to be improved for quality
Another reviewer recommended to remove figure 1. So, we removed it because everything is written in the text.

Reviewer 2 Report
TLS, the ectopic secondary lymphoid-like structures, are increasingly gaining attention due to their potential (demonstrated in some) benefits in prognosis or therapies for a variety of cancer types. The mechanistic investigation at the cellular or molecular levels, however, is still at the early stage. Indeed, even a general standard for TLS classification/categorization are still lacking.
This study conducts a statistical analysis of TLS in a small collection (>50) of breast patient samples, leading to the conclusions that BC molecular subtypes exhibit differential correlation with TLS; correlation between TLS and BMI(obesity) and some indication of TLS’ roles in angiogenesis. Adding to a number of earlier similar (but less extensive or with different emphasis) statistical studies of clinical samples, the results of these studies suggest interest leads for future molecular investigations.
The writing of this manuscript needs to be extensively polished. The current form is littered with numerous grammatic errors, incomplete sentences, misuse of phrases, etc., making the manuscript difficult to access.
A coherent summary of published statistical studies of TLS in BC, and how the current study lines up with them, perhaps in a table form, would be helpful for the readers to better grasp the main points of this work.
Author Response
RESPONSE TO REVIEWER 2
Dear Reviewer,
We would like to thank you for your time given for reviewing our paper. Your comments were valuable for us and helped us to improve the quality of the submission.
We are grateful for you positive appreciation related to our paper.
Please, find below our answers point by point to your comments. According to the instructions received from the editor, both for this letter and manuscript all the revisions will be highlighted in red.
With all our gratitude, in the name of manuscript authors,
Anca Maria Cimpean, MD, PhD, Hab.Dr., Full Professor of Histology, Pathologist
TLS, the ectopic secondary lymphoid-like structures, are increasingly gaining attention due to their potential (demonstrated in some) benefits in prognosis or therapies for a variety of cancer types. The mechanistic investigation at the cellular or molecular levels, however, is still at the early stage. Indeed, even a general standard for TLS classification/categorization are still lacking.
Thank you for your comment.
This study conducts a statistical analysis of TLS in a small collection (>50) of breast patient samples, leading to the conclusions that BC molecular subtypes exhibit differential correlation with TLS; correlation between TLS and BMI(obesity) and some indication of TLS’ roles in angiogenesis. Adding to several earlier similar (but less extensive or with different emphasis) statistical studies of clinical samples, the results of these studies suggest interest leads for future molecular investigations.
Yes, you are right, TLS impact on prognosis and therapy gradually increased.
The writing of this manuscript needs to be extensively polished. The current form is littered with numerous grammatic errors, incomplete sentences, misuse of phrases, etc., making the manuscript difficult to access.
English errors and spelling were extensively revised. You were right several typing and grammatical errors has been found during revision.
A coherent summary of published statistical studies of TLS in BC, and how the current study lines up with them, perhaps in a table form, would be helpful for the readers to better grasp the main points of this work.
Thank you for your suggestion. A table has been inserted at the end of discussion section thanks to your and to another reviewer suggestions. This will really be a welcome overview on the topic.

Reviewer 3 Report
In their study, Barb et al. evaluate the relationship between tertiary lymphoid structures and tumor-associated stromal blood vessels, and how they correlate with lymphovascular and perineural invasion across all molecular subtypes of breast cancer. While an interesting topic, this manuscript fails to adequately convey and interpret the findings. Elements of the manuscript and study can be strengthened as suggested below:
Abstract
· Define what BC is
· “…increased LIV and/or PnI and recurrence (p<0.001) excepting Luminal A type.” Please fix the grammar in this sentence.
· “Their antitumor immunity mediator’s activity seems to be strongly related to tumor aggressiveness and stromal blood vessels.” The relationship with the immune response is not properly captured in this manuscript and should therefore be removed.
Introduction
· The introduction needs to be cut down to the most relevant components pertinent to this research study. There is no need to describe the history of tertiary lymphoid structures.
· Please rephrase many of the sentences to be more concise and to improve the flow of the section. For example, “Tertiary lymphoid structures (TLS) are accumulations of lymphoid tissue organized as follicular-like structures into tissues other than the currently recognized lymphoid organs” can be simplified.
· Why were you only interested in lymphovascular and perineural invasion? What about other sites of recurrence?
Materials and Methods
· Some information is repeated between the different sub-sections. Please fix this and include information where it is most appropriate.
Results
· In section 3.1, I suggest presenting the distributions of tumor grade and BC subtypes in a table.
· In section 3.1, the percentages between Nottingham score and tumor grade appear inconsistent. It is my understanding that a Nottingham score of 3-5 represents a grade 1 tumor, a score of 6-7 represents a grade 2 tumor, and a score of 8-9 represents grade 3. As an example, you report that 3.77% of total BCs (53) are G1. This means that 3.77% of 53 = 2 tumors are G1. You also report that 1.88% of cases had a Nottingham score of 4 and 3.77% of cases had a Nottingham score of 5. I am assuming that the number of cases is still 53 here, so 1.88% of 53 + 3.77% of 53 = 3 cases that are grade 1 based on the Nottingham scores. The number of grade 1 tumors does not match between these two (2 vs. 3). Check all numbers or comment on the inconsistencies.
· Delete Figure 1 and see first comment. Also, for future figures, please make sure they are in a professional format.
· “Round or oval in shape well organized lymphoid follicle-like structures (Figure 2B)…” Be a little more descriptive to support Figure 2B
· Add arrows pointing to TLS in Figure 2 to make the images easier to interpret.
· Figure 2H caption says “Says, TSL…” correct TSL to TLS.
· Under section 3.2, second paragraph it says “About 5% of BC cases had intratumor…”. Is this 5% of all BC cases or of TLS+ cases? In the same sentence, it says “…while the resting 5%...”. Do you mean the rest of the 95% of cases? Please correct and clarify.
· “TLS close to adipose tissue were big in size and usually ill defined.” What do you mean by ill-defined? Elaborate on this.
· “These TLS type was highly vascularized compared to intra-tumor TLS or those found in between normal breast tissue and tumor area (Figure 2B).” Earlier you said TLSs weren't found in normal adjacent tissue, but this implies that TLSs were found adjacent to the tumor.
· You justify using BMI in the discussion section. This should be moved earlier to the section where you mention BMI. However, another metric for evaluating the effects of adipose tissue should also be used as BMI alone could potentially lead to misinterpreting the data.
· Please make Figure 3 more professional and only include the most relevant elements.
· “…and may be considered an independent negative prognostic factor which indirectly may favour lymphovascular invasion and metastases.” This statement is weak and lacks sufficient support.
· In section 3.3.3, you do not comment on the correlation with blood vessels as was done in the other sections.
· Table 1 needs to be reformatted to something that is easier to read and is more professional.
· No caption is required under Table 1
Discussion
· No true discussion is made, merely comparisons with current literature and a summary of the results. The implications of the findings should be discussed. It is not enough to say that all subtypes have been looked at. What are the remaining gaps in this field?
· Cancer immunotherapy is mentioned in the discussion, introduction, and simple summary but bears no relevance to this particular study.
· “In the present study, we proved that BMI significantly influenced TLS presence in BC.” Your results showed that Luminal A-BC cases showed significant correlation between TLS and BMI. The language in this statement should be more specific.
· “Our accurate TLS microscopic assessment highlighting TLS topographic and morphometric details is in line with Sawada findings and recommendations.” This statement is not adequately supported based on the results presented. Firstly, you do not confirm HEV protein expression. Additionally, Table 1 shows that TLS+ Luminal A correlates with perineural and lymphovascular invasion while TLS- Luminal A shows no significant correlation. This would suggest that TLS+ Luminal A patients would fare worse in the clinic due to the presence of these TLS. On the other hand, Sawada et al. shows that an HEV signature in TLS+ areas is associated with increased survival for BC patients, implying that TLS presence is a good thing. However, these two findings oppose each other.
· This manuscript would benefit from incorporating mechanistic and therapeutic studies. The novelty of this study is lost if you report that “To the best of our knowledge there are no studies in literature which report TLS percentage variability amongst all different molecular subtypes. There is only one published paper [51] where TLS presence was evaluated separately for each molecular subtypes, but it lacks Luminal B-HER2 type from analysis.”
Author Response
RESPONSE TO REVIEWER 3
We would like to thank you for your time given for reviewing our paper. Your comments were valuable for us and helped us to improve the quality of the submission.
Please, find below our answers point by point to your comments. According to the instructions received from the editor, both for this letter and manuscript all the revisions will be highlighted in red.
I would also like to mention that your review was one of the most valuable reviews of this paper and from my whole life of researcher (about 20 years) which gave to our paper another face. It is practically another paper, hopefully better based on your valuable suggestions. And for sure this will be my longest Response to reviewer since now!
With all our gratitude, in the name of manuscript authors,
Corresponding author,
Anca Maria Cimpean, MD, PhD, Hab.Dr., Full Professor of Histology, Pathologist
In their study, Barb et al. evaluate the relationship between tertiary lymphoid structures and tumor-associated stromal blood vessels, and how they correlate with lymphovascular and perineural invasion across all molecular subtypes of breast cancer. While an interesting topic, this manuscript fails to adequately convey and interpret the findings. Elements of the manuscript and study can be strengthened as suggested below:
Abstract
- Define what BC is
Your observation was pertinent, thank you. Due to a several reviews of the abstract for respecting words count, we were removed the first phrase where the term breast cancer is mentioned in full length before the use of its abbreviation- BC. So, as a conclusion, BC is abbreviation from Breast Cancer. This was useful to use because of a proper management of words count inside the abstract. In the revised version of the abstract, we fixed this issue, and we added the full-length breast cancer before the first use of abbreviation (BC). Please see the manuscript where all changes made to Abstract section were highlighted in red according to Instructions for authors.
- “…increased LIV and/or PnI and recurrence (p<0.001) excepting Luminal A type.” Please fix the grammar in this sentence.
We rephrase this part (please, see the manuscript) and hopefully now is easier to read and more concise, also.
- “Their antitumor immunity mediator’s activity seems to be strongly related to tumor aggressiveness and stromal blood vessels.” The relationship with the immune response is not properly captured in this manuscript and should therefore be removed.
You were right, the conclusion was not in the light of the study. Thus, we totally changed this phrase for a better reflection of our findings.
Introduction
- The introduction needs to be cut down to the most relevant components pertinent to this research study. There is no need to describe the history of tertiary lymphoid structures.
We thought present paper as having a high addressability from young researchers to senior researchers in case if it will be published. Our intention was not to make a history of TLS. We made an overview of disease where TLS were described first, pointing out that TLS are not specific for malignant diseases and, despite of their observation and early description they had pay no so much attention as they probably deserve. This overview was done not only in the context of TLS potential role in tumorigenesis but also related to the angiogenic (stimulation of new blood vessels development) potential of lymphoid tissues from different inflammatory, autoimmune and malignant disease. This is in line with our aim regarding the interplay in between TLS and stromal blood vessels (interrelation in between two stromal components with an impact on tumor progression). Thus, we consider that this overview would be helpful for young researchers who is not yet familiar with such structures.
We took into consideration your suggestions and we refined Introduction by removing years from the text. We agreed that years are not relevant, but the content is. So, we transformed a historical perspective into a scientific, professional one, highly relevant to our aim by rephrasing and shortening the actual content. Thank you!
- Please rephrase many of the sentences to be more concise and to improve the flow of the section. For example, “Tertiary lymphoid structures (TLS) are accumulations of lymphoid tissue organized as follicular-like structures into tissues other than the currently recognized lymphoid organs” can be simplified.
We rephrased it, thank you!
- Why were you only interested in lymphovascular and perineural invasion? What about other sites of recurrence?
We were not interested exclusively in lymphovascular and perineural invasion. Our database includes a unique archive of data composing of more than 60 clinico-pathological parameters individually assessed for each patient. To include all these parameters related to TLS in one article would be too much for a scientific report. We selected lymphovascular/perineurial invasion and recurrence due to several reasons in the context of our purpose of the study: (i) lymphatic/ blood vessels and nerves are components of BC tumor stroma as TLS are so, we may study interrelation in between several components of the stroma but having a prognostic impact and predicting survival (presence of invasion on microscopic specimens is considered a bad prognostic marker, for example) (ii) lymphovascular and perineurial invasion are quantified microscopically as TLS are, so an objective overview and credible interrelation would be assessed; (iii) lymphovascular and perineural invasion represents ways of local and distant tumor dissemination not sites of recurrence but recurrences is highly and partially dependent by lymphovascular and perineurial invasion. When we used recurrence, we referred to loco-regional recurrence as the most important site which may be properly managed and treat to avoid the tumor spread as distance metastasis and recurrences to metastatic sites far from the primary tumor.
Materials and Methods
- Some information is repeated between the different sub-sections. Please fix this and include information where it is most appropriate.
Materials and methods section was extensively revised. We hope that now nothing seems duplicated. Thank you!
Results
- In section 3.1, I suggest presenting the distributions of tumor grade and BC subtypes in a table.
- In section 3.1, the percentages between Nottingham score and tumor grade appear inconsistent. It is my understanding that a Nottingham score of 3-5 represents a grade 1 tumor, a score of 6-7 represents a grade 2 tumor, and a score of 8-9 represents grade 3. As an example, you report that 3.77% of total BCs (53) are G1. This means that 3.77% of 53 = 2 tumors are G1. You also report that 1.88% of cases had a Nottingham score of 4 and 3.77% of cases had a Nottingham score of 5. I am assuming that the number of cases is still 53 here, so 1.88% of 53 + 3.77% of 53 = 3 cases that are grade 1 based on the Nottingham scores. The number of grade 1 tumors does not match between these two (2 vs. 3). Check all numbers or comment on the inconsistencies.
Thank you very much for this comment but as a pathologist I did not really understand it. I must explain to you some issues. Most probably you are not pathologist and numbers and comments seems to you inconsistent. You associated in a non-pathological original manner Nottingham Prognostic Index with tumor grade differentiation which are TWO DIFFERENT ENTITIES in pathology. You were right, we have only 2 cases with G1 (as a pathologist we consider these as lucky patients!!!). The percentage of 3.77% is just a number coincidence in between G and NPI. Please find below G for cases you mentioned in this comment as NPI 4 and 5. Also please find below some explanatory details about difference in between G and NPI. Your differences of one case are based on your understanding of both G and NPI.
|
NPI |
G |
|
4 |
1 |
|
5 |
1 |
|
5 |
2 |
Yes, we have 3 cases together of 4 and 5 (as you correctly calculated) but not all must have G1. We have 2 with G1 (as we and you correctly calculated) and one with G2 because the final NPI is calculated as below. To other G2 cases correspond a higher NPI.
- Tumor differentiation grade (G) definition: Tumor grade describes how normal or abnormal cancer cells look under a microscope. https://www.cancer.gov/about-cancer/diagnosis-staging/diagnosis/tumor-grade#:~:text=But%20most%20tumors%20are%20graded,2%2C%203%2C%20or%204.&text=In%20grade%201%20tumors%2C%20the,4%20tumors%20look%20most%20abnormal
- NPI definition and calculation:
The Nottingham prognostic index (NPI) is used to determine prognosis following surgery for breast cancer.[1][2] Its value is calculated using three pathological criteria: the size of the tumour; the number of involved lymph nodes; and the grade of the tumour.[1] It is calculated to select patients for adjuvant treatment.
Calculation
The index is calculated using the formula:[3]
NPI = [0.2 x S] + N + G
Where:
- S is the size of the index lesion in centimetres[3]
- N is the node status: 0 nodes = 1, 1-3 nodes = 2, >3 nodes = 3[3]
- G is the grade of tumour: Grade I =1, Grade II =2, Grade III =3[3]
As it has be seen in this formula it is innapropiate to make correspondences in between G and NPI due to the high variability of tumor size and node status.
Interpretation[edit]
|
Score |
5-year survival[4] |
|
>/=2.0 to </=2.4 |
93% |
|
>2.4 to </=3.4 |
85% |
|
>3.4 to </=5.4 |
70% |
|
>5.4 |
50% |
- Delete Figure 1 and see first comment. Also, for future figures, please make sure they are in a professional format.
We deleted the figure 1. You were right is not so relevant once everything is writing in the text. When we added it, we assumed that is easily and quick to read but we do not want to be boring, so we removed it! Thank you!
I did not really understand how to define professional format for the figures but, I would like to mention that all pictures from this paper were in accordance with Instructions for authors from Cells journal. If they would not be, the handling editor (who check first these details) would send back the paper to the authors before sending it to reviewers and this did not happen.
- “Round or oval in shape well organized lymphoid follicle-like structures (Figure 2B)…” Be a little more descriptive to support Figure 2B·
Add arrows pointing to TLS in Figure 2 to make the images easier to interpret
This is as much as descriptive the picture 2 offers. This picture is used to point out morphologic heterogeneity related to the shape. I am not able to find synonymous for oval and round and thus I choosed to use coloured shapes. We added in picture 2 a blue oval shape to highlight the oval TLS and a yellow circle to highlighted round TLS.
- Figure 2H caption says “Says, TSL…” correct TSL to TLS.
Corrected, thank you!
- Under section 3.2, second paragraph it says “About 5% of BC cases had intratumor…”. Is this 5% of all BC cases or of TLS+ cases? In the same sentence, it says “…while the resting 5%...”. Do you mean the rest of the 95% of cases? Please correct and clarify.
Round or oval well organized lymphoid follicle-like structures (Figure 2B) were detected in 90% out of total 53 cases heterogeneously distributed inside TME (Figure 2C). About 5% out of total 53 BC cases had intratumor TLS (Figure 2G, J) while the remaining 5% out of total 53 showed TLS in the BC tumor microenvironment far from the tumor often in between stroma and adipose tissue surrounding mammary gland (Figure 2D, E).
- “TLS close to adipose tissue were big in size and usually ill defined.” What do you mean by ill-defined? Elaborate on this.
I elaborated it and changed it! In pathology, ill-defined=not sharply delineated from surrounding tissue.
- “These TLS type was highly vascularized compared to intra-tumor TLS or those found in between normal breast tissue and tumor area (Figure 2B).” Earlier you said TLSs weren't found in normal adjacent tissue, but this implies that TLSs were found adjacent to the tumor.
I found this very interesting comment. I am sure that in Figure 2A you recognized Terminal Ductal Lobular Unit (TDLU) the morpho functional unit specific for the normal breast. As you probably see in figure 2A TLSs weren't found in normal adjacent tissue
In the upper left corner of figure 2B, I am sure that you recognize malignant tissue based on pathologic microscopic criteria and on opposite lower right corner the normal adjacent tissue (I mean adjacent normal breast tissue) IN between these two corners it is easily recognizable tumor stroma and inside it highlighted TLS in between tumor (upper left) and normal (lower right) adjacent tissue. I would like to mention that inside normal is not like adjacent to normal. For a better understanding for a wide spectrum of medical specialities we changed in the text in with inside and adjacent to close to…normal breast.
Additionally, in the light of your suggestions for other pictures, we considered appropriate to insert also in previous figure 2G (actually figure 1G) arrows and numbers pointed TLS (1) and tumor tissue (2) for a better understanding of what it means intratumor TLS. Thank you!
- You justify using BMI in the discussion section. This should be moved earlier to the section where you mention BMI. However, another metric for evaluating the effects of adipose tissue should also be used as BMI alone could potentially lead to misinterpreting the data.
One of the purposes of the present study was to correlate microscopic TLS presence to clinical parameters. It is widely accepted that adipose tissue is a trigger for an increased inflammatory response in both benign and pathologic conditions. Thus, we choosed BMI. You were very elusive related to another metric for evaluating adipose tissue because you did not recommend us a specific metric punctually, you did a general comment on this. But, we already assessed another metric as you called it. In this context I would like to tell you that one part of our big project in BC stroma assessment is the evaluation of adipose tissue area and vascularization impact on other stromal components (including TLS) and malignant cells behaviour. The volume data from this new research directions was compiled in to other two distinct papers which were already submitted for publication to other journals so, we are not allowed to repeat them in the present paper because of the rules of scientific research results publication.
The following phrase has been moved from Discussion to results section where you suggested.
No data about TLS related to Luminal A, B or Luminal B-HER2 molecular subtypes are available in this moment nor their relationship to BMI, age, recurrence or lymphovascular and perineural invasion.
- Please make Figure 3 more professional and only include the most relevant elements.
Figure 3 was automatically generated by statistical analysis software and has a pre-defined format which does not allowed us to change any result or format.
If professional means image quality, then we checked it and image quality of 330dpi is in accordance to instructions for authors.
- “…and may be considered an independent negative prognostic factor which indirectly may favour lymphovascular invasion and metastases.” This statement is weak and lacks sufficient support.
Ok, Removed!
- In section 3.3.3, you do not comment on the correlation with blood vessels as was done in the other sections.
Yes, we did not comment because no relevant findings has been observed. I added the following phrase to the end of 3.3.3. subsection
No significant and relevant findings had been observed related to both type of stromal blood vessels.
- Table 1 needs to be reformatted to something that is easier to read and is more professional.
There is the simplest form to summarize such so many data. We checked for more than 3 other different formats and we choose this because it is the best reflection of comparative assessment in between BC molecular subgroups.
Other reviewers truly appreciate the table format and appreciate its conciseness.
- No caption is required under Table 1
Table 1 has no caption, just Legend required by the editor.
Discussion
- No true discussion is made, merely comparisons with current literature and a summary of the results.
All your comments and suggestions for the present paper were so valuable and useful to us but, with all my gratitude to your time and work, I would like to tell you that we are not at our first paper. Somehow, your comment related to discussion section was not such an elegant and objective one. It sounds as a critical comment done for the beginners who did not write any article till now. I would really like to mention that I am handling editor to BMC and MDPI groups and Guest Editor to Frontiers and other several prestigious journals groups and for sure that they would not select me to do this if I would not pass some preliminary steps toward this, including several papers published there and elsewhere (about 200 totally in web of science). I would like to ask you to keep a balanced evaluation of this paper. No other reviewer of this paper (other 3!) did mention any trouble to this section. The discussions were made by comparisons with current literature and a summary of the results. We presented literature data in context of the present study. There were critically discussed our findings related to previous existing ones. There are no missing parts of our results remaining without discussion in this section. Maybe you should read again carefully this part.
The implications of the findings should be discussed. It is not enough to say that all subtypes have been looked at. What are the remaining gaps in this field?
The implication of the findings was reviewed by the authors. They are several times discussed! The gaps were mentioned in the text. The lack of TLS inclusion as a criterion of routine diagnostic histopathology is one of the gap and was mentioned in the text. TLS involvement in metastasis is still a gap and it is mentioned in the paper several times. Interplay in between stromal blood vessels and TLS is not clear…TLS and adipose tissue interrelation is also a gap in this research. All of these gaps were discussed in this paper.
- Cancer immunotherapy is mentioned in the discussion, introduction, and simple summary but bears no relevance to this particular study.
Cancer immunotherapy mention is not a mistake and will not be removed from the text because is inserted here to highlight that TLS assessment should be mandatory in the future because is one of the most promising targets for immunotherapy in cancer by its cellular content.
- “In the present study, we proved that BMI significantly influenced TLS presence in BC.” Your results showed that Luminal A-BC cases showed significant correlation between TLS and BMI. The language in this statement should be more specific.
We rephrased it as:
In the present study we proved that BMI significantly influenced TLS presence in some subtypes of BC , specifically for Luminal A subtype.
- “Our accurate TLS microscopic assessment highlighting TLS topographic and morphometric details is in line with Sawada findings and recommendations.” This statement is not adequately supported based on the results presented. Firstly, you do not confirm HEV protein expression. Additionally, Table 1 shows that TLS+ Luminal A correlates with perineural and lymphovascular invasion while TLS- Luminal A shows no significant correlation. This would suggest that TLS+ Luminal A patients would fare worse in the clinic due to the presence of these TLS. On the other hand, Sawada et al. shows that an HEV signature in TLS+ areas is associated with increased survival for BC patients, implying that TLS presence is a good thing. However, these two findings oppose each other.
You were right with your observation. The statement was removed. Thank you.
- This manuscript would benefit from incorporating mechanistic and therapeutic studies.
They were incorporated based on your suggestions and another reviewer suggestion. Thank you.
The novelty of this study is lost if you report that “To the best of our knowledge there are no studies in literature which report TLS percentage variability amongst all different molecular subtypes. There is only one published paper [51] where TLS presence was evaluated separately for each molecular subtypes, but it lacks Luminal B-HER2 type from analysis .This phrases were removed.

Reviewer 4 Report
The work of Barb and colleagues provides a comprehensive overview of the current literature on tertiary Lymphoid structures (TLS) and their role in Breast cancer. The introduction is very informative and detailed.
However, the provided experimental work failed to address the author's main aims, and together with substantial missing information and validation analyses preclude the possibility of asking for a revision. The entire work should be completely revised according to a more punctual statement of the work's main aims and the experimental approaches to answer each aim.
Main points
- The main aim of the work is not completely clarified at the end of the introduction "The aim of this work includes the impact of TLS presence on tumor stroma blood vessels type". It is not clear which experiment addresses this aim since no images are provided with a spatial correlation between TLS and blood vessels.
- Quantitation of the amount of TLS in each TLS+ patient is not provided.
- The IHC images are not provided at all. How can we evaluate the author's staining and quantitation of blood vessels? Also, I would recommend using the CD31 marker to stain mature endothelial cells in combination with alpha-SMA. Quantification of blood vessels with statistics is also required for each BC subtype (data are not shown).
- The CD45+ staining is required to confirm the presence of immune cells in TLS.
Minor points
- The authors identified a significant difference in the TLS presence among different BC subtypes. This is an interesting finding. However, the figures provided are very poorly representative. In the figure legend, the breast cancer type is not even mentioned, as well as the type of staining (Figure 2). Please provide an overview of different BC types analyzed with representative H&E images.
- A more detailed description of TLS is foreseen. For example, 253 line states that TLS from different areas (stroma/adipose tissue and adipose tissue/tumors) are different. Please add more morphological information to support this notion.
- Also, it would be helpful to add arrows on images to point out TLS (for example in panels 2A and 2B).
- The authors stated that TLS are not present in normal breast tissue (line 235) but this is contradicted in line 256.
Author Response
RESPONSE TO REVIEWER 4
We would like to thank you for your time given for reviewing our paper. Your comments were valuable for us and helped us to improve the quality of the submission.
Please, find below our answers point by point to your comments. According to the instructions received from the editor, both for this letter and manuscript all the revisions will be highlighted in red.
With all our gratitude, in the name of manuscript authors,
Corresponding author,
Anca Maria Cimpean, MD, PhD, Hab.Dr., Full Professor of Histology, Pathologist
The work of Barb and colleagues provides a comprehensive overview of the current literature on tertiary Lymphoid structures (TLS) and their role in Breast cancer. The introduction is very informative and detailed.
Thank you so much for your appreciation. Based on suggestions of Reviewer 3, I would like to inform you that Introduction was revised but it remained informative and detailed.
However, the provided experimental work failed to address the author's main aims, and together with substantial missing information and validation analyses preclude the possibility of asking for a revision. The entire work should be completely revised according to a more punctual statement of the work's main aims and the experimental approaches to answer each aim.
Thanks for your opinion. The paper was extensively revised, and your requests were resolved. The aim was rephrase for a better understanding.
Main points
- The main aim of the work is not completely clarified at the end of the introduction "The aim of this work includes the impact of TLS presence on tumor stroma blood vessels type".
The aim of the work was revised. In this moment it sounds as follow:
We aim to study the interplay in between TLS and tumor stroma blood vessels (immature-CD34+/SMA -, versus mature- CD34+/SMA+), to find if this interrelation is BC subtype specific and may influence lymphovascular and perineurial invasion and recurrence, also.
It is not clear which experiment addresses this aim since no images are provided with a spatial correlation between TLS and blood vessels.
Most probably you had a technical problem with images upload. The images are present and inserted in the manuscript at Results section and there are very well seen stromal blood vessels and intra TLS blood vessels stained by immunohistochemistry. On these images spatial correlation in between stromal blood vessels, TLS, normal and malignant tissue may be observed.
TLS were pointed by quadrans while vessels by arrows.
- Quantitation of the amount of TLS in each TLS+ patient is not provided.
TLS amount was quantified for each case, otherwise we would not be able to perform statistics. The number of TLS varied in between 1 to 3 per whole tissue biopsies. This statement was added in the text. Thank you!
- The IHC images are not provided at all.
Dear reviewer 4, I do really concern about the objective evaluation of this paper while you stated above that IHC images are not provided at all. I understand that the images were not uploaded on your manuscript (I do really hope that this is the problem).
The IHC images WERE provided in Figure 2 (Figure 1 in revised version). We added in the figure legend the IHC staining type. It is a double stain for CD34 (endothelial cells) and SMA (smooth muscle perivascular cells). Based on this staining we aim to identify stromal and intra TLS vessels of immature (CD34+/SMA-) and mature (CD34+/SMA+) type. This provided us an accurate spatial distribution of blood vessels closely related to TLS.
As a conclusion, something wrong happened to your manuscript for revision. I would advise you to ask to academic editor about sending you by email the version with IHC images included, otherwise I has to ask to academic reviewer for more reviewer’s opinion due to your contradicting statements from this review which may imply the objective evaluation of the paper.
How can we evaluate the author's staining and quantitation of blood vessels?
This is stated in the Material and Methods section from line 154 to line 162 as it follows:
From this digital library the slides were uploaded on QuPath version 0.4.2., an open-source platform for bioimage analysis of microscopic slides where they were analyzed by using integrated software and its extensions as Fiji and Vascular Analysis. For each case we selected three stromal areas with the highest density of stromal blood vessels close to tumor areas and TLS (where they are present) by using QuPath annotation tools. We differentially counted at X400 magnification the mature (CD34+/SMA+) and immature (CD34+/SMA-) in the same selected area. An average of mature and immature stromal blood vessels from those three areas were used for our purpose.
Also, I would recommend using the CD31 marker to stain mature endothelial cells in combination with alpha-SMA. Quantification of blood vessels with statistics is also required for each BC subtype (data are not shown).
Our purpose was not to quantify mature endothelial cells but mature tumor blood vessels (having SMA positive perivascular cells) which is highly different. CD31 is an adhesion molecule which may be missed in angiogenic endothelial cells due to heterogeneity of their interconnection during angiogenic process). It is not continuously expressed during tumor angiogenesis and thus we may lack some blood vessels development steps. CD34 is expressed inside endothelial cells from the beginning of development till the end of life. CD34 was the best choice for our purpose.
- The CD45+ staining is required to confirm the presence of immune cells in TLS.
CD45 may be required but is not mandatory since histologists and pathologists (I have both specialities), are able to recognize a lymphoid structure on routine evaluation. Thus in this study we did not use CD45 because is not relevant.
Minor points
- The authors identified a significant difference in the TLS presence among different BC subtypes. This is an interesting finding. However, the figures provided are very poorly representative.
I apologize but I am really confused now. Before you stated that IHC images are not provided. Now you address to these images because Figure 2 contains a representative collage of CD34/SMA doublestain IHC pictures taken from our cases specimens. You did not mentioned why the pictures are poorly representative? We consider that are not poorly representative , they are highly representative due to their content which includes both structures aiming this study: TLS and stromal blood vessels together with representative areas of tumor tissue and
In the figure legend, the breast cancer type is not even mentioned, as well as the type of staining (Figure 2).
`Conventional type of BC as ductal invasive or lobular invasive is established on H&E staining. We did it for diagnostic purpose but this does not help of our research purpose. We aim to study BC molecular subtypes related to TLS not conventional histopathologic types.
We added staining in the legend of figure.
Please provide an overview of different BC types analyzed with representative H&E images.
Molecular types of breast cancer which we are interested in are not established on H&E stain. They are classified based on IHC staining for estrogen and progesterone receptors (ER and PR), Ki 67 and HER2 oncoprotein. In the text there is already stated an overview about BC molecular subtypes included in this study.
- A more detailed description of TLS is foreseen. For example, 253 line states that TLS from different areas (stroma/adipose tissue and adipose tissue/tumors) are different. Please add more morphological information to support this notion.
A detailed description containing peculiarities is already present in the text
- Also, it would be helpful to add arrows on images to point out TLS (for example in panels 2A and 2B).
There were added. Figure 2 became figure 1 due to the remove of figure 1 according to the suggestion of reviewers 4 and 1
- The authors stated that TLS are not present in normal breast tissue (line 235) but this is contradicted in line 256.
I found this very interesting comment. I am sure that in Figure 2A you recognized Terminal Ductal Lobular Unit (TDLU) the morpho functional unit specific for the normal breast. As you probably see in figure 2A TLSs weren't found in normal adjacent tissue
In the upper left corner of figure 2B, I am sure that you recognize malignant tissue based on pathologic microscopic criteria and on opposite lower right corner the normal adjacent tissue (I mean adjacent normal breast tissue) IN between these two corners it is easily recognizable tumor stroma and inside it highlighted TLS in between tumor (upper left) and normal (lower right) adjacent tissue. I would like to mention that inside normal is not like adjacent to normal. For a better understanding for a wide spectrum of medical specialities we changed in the text in with inside and adjacent to close to…normal breast.

Round 2
Reviewer 2 Report
Authors have addressed my concerns on the earlier version adequately.
Reviewer 4 Report
No more comments